# Differences and Similarities in Adaptive Functioning between Children with Autism Spectrum Disorder and Williams–Beuren Syndrome: A Longitudinal Study

**DOI:** 10.3390/genes13071266

**Published:** 2022-07-16

**Authors:** Paolo Alfieri, Francesco Scibelli, Federica Alice Maria Montanaro, Maria Cristina Digilio, Lucilla Ravà, Giovanni Valeri, Stefano Vicari

**Affiliations:** 1Child and Adolescent Neuropsychiatry Unit, Department of Neuroscience, Bambino Gesù Children’s Hospital, IRCCS, 00153 Rome, Italy; francescoscibelli@gmail.com (F.S.); federica.montanaro@opbg.net (F.A.M.M.); giovanni.valeri@opbg.net (G.V.); stefano.vicari@opbg.net (S.V.); 2Medical Genetics Unit, Bambino Gesù Children’s Hospital, IRCCS, 00153 Rome, Italy; mcristina.digilio@opbg.net; 3Epidemiology Institute, Bambino Gesù Children’s Hospital, IRCCS, 00165 Rome, Italy; lucilla.rava@opbg.net; 4Department of Life Sciences and Public Health, Università Cattolica del Sacro Cuore, 00168 Rome, Italy

**Keywords:** adaptive functioning, cognitive functioning, rare genetic syndrome, behavioral phenotype, intellectual disabilities, ASD, 7q11.23

## Abstract

Background: The last decade has seen a growing number of comparative studies on adaptive profiles between individuals with autism spectrum disorder (ASD) and Williams–Beuren syndrome (WBS), showing shared and syndrome-specific adaptive trajectories. Studies have revealed similarities in global adaptive profiles across conditions, while some differences have been found in preschoolers on the specific sub-domains of communication and socialization. However, the majority of studies that have focused on the differences in adaptive functioning across these two conditions used a cross-sectional design. To the best of our knowledge, there are no studies exploring the differences and similarities of adaptive functioning over time. Methods: We compared longitudinal data of adaptive functioning measured by Vineland Adaptive Behavior Scales (VABS) between two samples of children and adolescents with ASD and WBS, matched for chronological age and cognitive/developmental level at the time of the first evaluation. Results and Conclusions: We did not find any difference on the global adaptive level, both at the first evaluation and over time. However, significant differences emerged on the socialization and communication levels at the time of recruitment. Longitudinal data show that only the socialization domain remains different over time, with individuals with WBS having better functioning than those with ASD. The results on shared and distinct patterns of adaptive functioning between disorders are discussed from a developmental perspective, thus contributing to the implementation of age-specific interventions.

## 1. Introduction

Adaptive functioning is defined as “the collection of conceptual, social, and practical skills that are learned and performed by people in their everyday lives” [1]. Adaptive functioning represents the typical performance usually reached by people, rather than being a measure of “actual ability” [2]. Research has showed that adaptive functioning is an essential long-term outcome in individuals with several developmental disabilities.

Autism spectrum disorder (ASD) is a group of heterogeneous neurodevelopmental disorders that share socio-communicative deficits and restricted, repetitive, and stereotyped patterns of behaviors and interests [3,4]. The estimated prevalence in the USA is 1:54 [5], while in Italy, it is slightly lower (1:87) [6]. ASD is considered an etiologically heterogeneous condition, due to the interplay between different environmental and genetic features [4,7].

Williams–Beuren syndrome (WBS) is a rare genetic syndrome caused by a continuous hemizygous microdeletion on chromosome 7q11.23. The deletion usually ranges from 1.55 Mb in 95% of cases to 1.84Mb in 5% of cases [8,9]. The estimated prevalence is 1:7500 [10]. Individuals with WBS share a common neurobehavioral phenotype that usually includes global developmental delay, mild-to-severe intellectual disability (ID), limited adaptive skills, major impairment in visuospatial abilities, expressive language skills relatively preserved if compared to comprehension skills and, finally, social communication difficulties [11,12,13,14]. Individuals with WBS are well known for their hyper-sociable behavior, which has led them to be considered as having an opposite social behavior to individuals with ASD.

Several studies have explored adaptive profiles in individuals with ASD or WBS [15,16,17,18,19,20,21,22,23,24]. Since the last decade, a growing number of longitudinal studies on adaptive functioning in ASD are available [25,26,27,28,29,30,31]. On the contrary, longitudinal data on adaptive functioning of individuals with WBS are more limited [32,33].

Broadly speaking, longitudinal studies on adaptive profiles of children with ASD showed a negative relationship between age and adaptive functioning [26,34], with ASD adaptive skills not keeping pace with typical adaptive growth. Notwithstanding, some researchers have highlighted that improvements in adaptive functioning are approximately achievable in 20% of diagnosed children, even though this ameliorating trajectory is probably limited to higher-functioning individuals. Studies on specific adaptive domains (communication, daily living skills, socialization, and motor skills) show that children acquire gains in daily living skills [31]. 

Longitudinal data on adaptive behavior in children with WBS are scarce [32,33]. The few available studies show a stable or declining trajectory of adaptive functioning after a two-/three-year period of time [33], while a general decreasing trend over time has been found in adolescents and adults, with the exception of the socialization domain [33]. Given that cognitive impairment is notoriously common in individuals with WBS, it is worth noting that these results are mainly related to individuals with ID.

In recent years, several cross-syndrome studies have been conducted in order to compare neurobehavioral profiles of children with different conditions or disorders [35,36,37,38]. Some cross-sectional studies have been carried out in order to address shared and distinct features of adaptive profiles between children with ASD and WBS [37,38], thus providing a snapshot of similarities and differences across conditions. The available cross-sectional studies on comparison between adaptive profiles of children with WBS and ASD have showed a lack of difference in global adaptive profiles across the two conditions [37,38], while some differences have been revealed on specific domains of communication or socialization in preschoolers [37,38]. To the best of our knowledge, there are no studies comparing longitudinal data of adaptive behavior across ASD and WBS.

To summarize, a growing number of studies provide information on adaptive trajectories of individuals with ASD, while studies on individuals with WBS are still lacking. More recently, the few studies that focused on differences between adaptive functioning across these two conditions mainly used cross-sectional designs. Cross-sectional studies are useful to better distinguish between syndromes; however, if not from a longitudinal perspective, they cannot provide a picture of how disorders change over time. To the best of our knowledge, there are no studies exploring differences and similarities of adaptive functioning over time, which is essential for a deeper definition of neurodevelopmental disorders and then for timely interventions. 

The main aim of this research is to contribute to filling this gap in research comparing longitudinal data of adaptive functioning between two samples of children and adolescents with ASD and WBD, matched for chronological age and cognitive/developmental level. On the basis of the abovementioned research, our main hypothesis is that children with ASD and WBS will have similar global adaptive functioning over time. Furthermore, we do not expect to find differences on all the investigated domains, except for socialization. This last assumption is based on the stable trajectory of socialization in WBS persisting in adulthood [33], while in ASD, the improvements in communication results were limited over time [31].

## 2. Materials and Methods

### 2.1. Participants

In total, 48 children and adolescents, 24 with ASD and 24 with WBS, were recruited at the Child and Adolescent Psychiatry Unit of Bambino Gesù Children Hospital. Individuals of the two groups were matched for chronological age and intelligent quotient (IQ)/developmental quotient (DQ) (*p*-values > 0.05). All information on age and gender is displayed in Table 1. 

All children had a normal standard chromosome analysis. Molecular testing for Fragile X syndrome was normal in all ASD patients. Microarray analysis in patients with WBS showed the classical 1.6 Mb heterozygous microdeletion in 7q11.23 in 22 cases with a larger 3.8 Mb microdeletion (from 72,726,578 to 76,583,962 in hg19) in 1 case. One patient with WBS and normal microarray analysis had a de novo heterozygous pathogenic variant (c.205delG; p.Ala71Argfrter51) in *ELN* (Elastin) gene mapping inside the critical 7q11.23 region. Microarray analysis was normal in 23 patients with ASD. In one patient, 3 copy number variants (CNVs) of unknown significance segregating from unaffected parents were detected. In one ASD patient, exome sequencing using a Clinical Exome Panel analysed on NextSeq550 showed four heterozygous variants classified as Vous (variants of unknown significance) in autosomal recessive genes, interpreted as not diagnostic for the patient.

All parents signed an informed consent form for research purposes. The study was approved by the Ethical Committee of our hospital (number of protocols: 1125).

### 2.2. Molecular Analysis

Exome sequencing was performed according to local protocols, namely, standard chromosome analysis on peripheral blood lymphocytes. Microdeletions were identified by array-CGH (Agilent, Santa Clara CA, USA) or SNP-array (Beadchip 850K, Illumina, San Diego, CA, USA) platforms at a resolution of 100 kb. Comprehensive open reading frame/splice site mutational analysis of clinical exomes were conducted through the Twist Human Core Exome Kit and sequenced on the Illumina NovaSeq6000 Platform. Lastly, an amplification study of the CGG repeats of the FMR1 gene for X fragile syndrome was carried out using RP-PCR. Further information is available upon request.

### 2.3. Assessment Tools

#### 2.3.1. Adaptive Level

Assessment of adaptive functioning has been evaluated by means of the Vineland Adaptive Behavior Scale (VABS). Vineland systems are considered the “gold standard” tool for adaptive functioning evaluation [39]; furthermore, VABS have commonly been used in studies involving both populations of children with ASD [15,40,41] and WBS [16,17,18]. The last VABS edition is VABS-III [2]; however, this form is still not available in Italian and VABS-II was also unavailable during the period in which data collection started; therefore, the first edition was used. VABS is a caregiver semi-structured interview and yields four domain scores: communication, daily living skills, socialization, and motor skills. Adaptive Behavior Composite (ABC) score is calculated by totaling the three (or four, in the case of the motor skills domain evaluation according to age) domain scores. As the motor skills domain was only administered to children younger than 6 years of age, we did not include it in the analysis.

#### 2.3.2. Cognitive/Developmental Level

Cognitive/developmental level has been assessed through the administration of appropriate tools, which have been selected on the basis of chronological age, developmental level, and language abilities. Then, the following scales have been used:Griffiths Mental Development Scales—Extended Revised (GMDS-ER) [42], a tool used to assess the level of development in children from birth to 8 years. GMDS-ER provides an overall score of developmental level (DQ), which has been used to compare the youngest children to the other patients included in our sample.The Leiter-R [43], a non-verbal cognitive tool administered to people from 2 years to 20 years and 11 months of age. This test does not require language abilities. In this study, the nonverbal brief intelligence quotient (hereinafter referred to as IQ) has been used for comparisons.Raven’s Colored Progressive Matrices (R-CPM) [44], a standardized test for children between 5 and 11 years of age. The assessment comprises 36 items and participants are asked to complete a missing piece out of six/eight options. This test provides a performance percentile and a corresponding intelligent quotient (hereinafter referred to as IQ).WISC–III [45] has been administered to children aged between 6 years and 16 years and 11 months. A Full-Scale Intelligent Quotient (hereinafter referred to as IQ) is available. Given the extended time of evaluations, the latest edition of WISC-IV [46] has been used.

IQ/DQ are all expressed in standard score (M = 100; SD = 15).

### 2.4. Procedure

Data for this study were retrospectively collected from a database including 75 patients with WBS referred to our hospital for routinary psychiatric evaluation. Twenty-four children with WBS met our inclusion criteria. Subsequently, these 24 WBS children were matched for chronological age and IQ/DQ with 24 children with ASD, who were selected from a database of 2345 ASD children. Patients were assessed in the time period between 2008 and 2020. 

Inclusion criteria:A clinical diagnosis of WBS confirmed positive fluorescent in situ hybridization test or Array-CGH;A clinical diagnosis of “Autistic Disorder”, “Asperger’s Disorder”, and “Pervasive Developmental Disorders Not Otherwise Specified” based on Diagnostic and Statistical Manual of Mental Disorders, 4^th^, text revision criteria (DSM IV-TR) [47], or ASD based on Diagnostic and Statistical Manual of Mental Disorders-5 criteria (DSM 5) [3] and confirmed by gold standard assessment tools: Autism Diagnostic Observation Schedule-Generic (ADOS-G) [48], Autism Diagnostic Observation Schedule, Second Edition (ADOS-2) [49] and Autism Diagnostic Interview—Revised (ADI-R) [50].Age between 2 and 17.11 years.At least two assessments of adaptive profile by means of VABS [51].Assessment of cognitive/developmental level by means of appropriate developmental tools during the first evaluation (Time 0).

Exclusion criteria included the presence of a non-corrected visual or hearing impairment and/or the presence of a non-controlled severe medical condition. 

Cognitive/developmental tools were distributed as follows in WBS patients: 10 Leiter-r, 10 GMDS, 2 WISC III, 1 WISC-IV, and 1 R-CPM. In the ASD subjects, tools were administered as follows: 12 Leiter-r, 12 GMDS.

Assessment was conducted by an interdisciplinary team composed of child psychiatrists and psychologists. Diagnoses were based on clinical observations, developmental evaluation, standardized tests, and parent interviews.

As data were collected from routine psychiatric evaluations, children were differently assessed over time. After the first evaluation (T0) all the children with WBS were re-evaluated at T1 after a mean (M) of 26.79 months (standard deviation, SD: 16.79). At T2, 12 children with WBS were re-evaluated after an M of 18.83 months (SD:5.77). At T3, only 2 children with WBS were re-evaluated after an M of 46 months (SD: 48.08). Concerning children with ASD, after T0, all the children were re-evaluated at T1 after an M of 8.95 months (SD: 3.81). At T2, 21 children were re-evaluated after an M of 12.19 months (SD: 9.46). At T3, 12 children with WBS were re-evaluated after an M of 20.75 months (SD: 28.70). Seven children with ASD were also re-evaluated at a fourth time point after 14.14 months (SD: 4.60). Collectively, we obtained 62 evaluations for the WBS group and 88 for the ASD group.

Even though our data were collected in routine visits, they provided good coverage in both the ASD and WBS samples across the developmental period spanning the age range from 2 to 16.11 years (see Figure 1 for a graphical representation of data distribution in our samples).

## 3. Statistical Analysis

Demographic, clinical, and cognitive characteristics at baseline were described with counts, proportions, M, and SD, and were compared between the ASD and WBS groups through chi square, Fisher’s exact, and Student’s T-test. 

Due to the varying ages of children included in the study, VABS scores at each time point were expressed as the ratio of equivalent age and chronological age (REC). 

Longitudinal trajectories on adaptive behavior in ASD and WBS children were estimated through multilevel mixed linear regression models. Such models allow us to study time-repeated measures with different numbers of observations for each subject. Considering each VABS domain (communication, daily living skills, socialization) and the total score (ABC) as separate dependent variables, mixed-effect models were fitted by including diagnosis group, chronological age (linear and quadratic effects), and an interaction term between diagnosis and age; random intercept and random intercept and slope model were considered and the best fitting models for each outcome were determined by likelihood ratio test (LRT).

Results were considered statistically significant for two-sided *p*-value < 0.005. Statistical analyses were performed using Stata 17.1.

## 4. Results

At time T0, VABS-ABC scores and VABS domains were less than one for both WBS and ASD, with WBS children always performing slightly better (Table 2). Differences in REC score between ASD and WBS were only statistically significant for communication and socialization VABS domains (communication *p* = 0.021; socialization *p* = 0.029), with higher scores in WBS children.

Several mixed-effect models were fitted to communication, daily living skills, socialization, and ABC data (Table 3). For each domain, the best fitting models were those including random intercepts and linear trend for age. Focusing on comparisons between groups, we did not find statistically significant differences between ASD and WBS on daily living skills and on ABC scores, while an approaching—but not reaching—significance result was observed in communication (*p* = 0.055), where ASD performed worse than WBS children. The only statistically significant difference emerged in the socialization domain, where WBS obtained higher scores than ASD individuals. 

Without considering the diagnosis, we observed a negative relation between VABS-ABC and age, which means that there was a dramatic decreasing trajectory in adaptive functioning in both groups of children over time *(*Table 3, Figure 2d). 

In total, the communication and daily living skills trajectories showed a statistically significant decreasing trend with age (Figure 2a,b,d). Regarding the socialization domain, the adaptive trajectory decreased with age, with WBS children always showing better performance than ASD children (Figure 2c). 

## 5. Discussion

The main aim of this research was to compare longitudinal data of adaptive functioning between children and adolescents with ASD and WBS. Cross-sectional results confirm data previously found on adaptive functioning comparisons [37,38]. Individuals with ASD and WBS did not differ on global adaptive behavior, while some differences were detected in the socialization and communication domains, with children with WBS having a higher level of functioning in both. 

Longitudinal results showed that all the investigated domains in children with WBS and ASD have a descending trajectory, including global adaptive functioning. Thus, age seems to have a strong decreasing effect on all domains of adaptive functioning, confirming previous data reported in studies on the two separate conditions. 

Between-groups comparisons of VABS domains revealed some specific differences. While the two groups did not differ on daily living skills and ABC, socialization seemed to be significantly better in children with WBS over time. This result is consistent with previous cross-sectional studies on cross-syndrome adaptive comparisons [37], thus confirming that the two groups mainly differ in socialization over the years, not only at preschool age.

Notably, communication results only bordered on significant, which could mean that communication differences found in preschoolers in previous cross-sectional studies [38] were not maintained over time. On the other hand, since the difference was fairly close to significant, more data are needed before we can draw any conclusions. 

This study led to some considerations. First, it corroborates the hypothesis that the difference in communication between the two samples observed at preschool age tends to reduce over time. Previous studies have shown that the ability to communicate in children with WBS is essentially due to their expressive skills, while their receptive and pragmatic abilities seem to be reduced [13,14,38]. Difficulties in receptive and pragmatic abilities make it hard to keep pace with typical developmental communication demands, resulting in differences between WBS and ASD observed in the first years of life tending to disappear over time. Second, deficits in communication in individuals with WBS should be treated with specific interventions on pragmatic and receptive abilities in a wider period of life and not only when they are very young. Third, it is plausible that the socialization domain is “pulled up” by the enhanced “social motivation” of individuals with WBS. Indeed, previous studies have shown that individuals with WBS have difficulties in interpreting socio-perceptual cues [52] and problems in maintaining peer relations [53]; it is plausible that the main differences in the socialization domain between the two samples are more likely due to the hyper-sociability that allows individuals with WBS to be engaged and interested in people than the complex social skills required during adolescence. Finally, the absence of differences between samples in global adaptive functioning over the years should cast light on how debilitating WBS can be. Usually, WBS is considered “less severe” than ASD. Our results demonstrate that global adaptive functioning in these two different conditions is more similar than previously thought, even over time. The perception of different severity should be attributed to other features, such as hyper/hypo-sociability and the presence/absence of severe problem behaviors in the two conditions. The impact of ASD or WBS on adaptive functioning seems not to differ, with both conditions having a similar descending trajectory. 

The genetic basis of similarities and differences in the two groups of patients cannot, to date, be explained. In fact, while the WBS cohort is genetically homogeneous with the involvement of rearrangements in the common 7q11.23 chromosomal region, the etiology of ASD can be linked to different single and multiple chromosomal loci, although the specific diagnosis is often unidentifiable, as occurred in the patients in this study. Further genetics studies and comparisons are required before drawing any conclusions. 

This study has several limitations. Firstly, even though this is the first study to provide a longitudinal comparison between ASD and WBS, our data have been collected at heterogeneous points in the history of children’s disorders. Future studies should use inception cohorts. Secondly, given that data were collected in clinical settings during routinized medical consultations, a convenience sampling method was used in this study, thus threatening the external validity of our research. Another important limitation of our study is that the cognitive/developmental level of children involved was always below average, thus excluding the possibility of extending these results to individuals with ASD or WBS with higher functioning. Additionally, cognitive profiles have been assessed using different tools, which, independently from the outcome (IQ), may not have the same level of complexity. Another limitation is that WBS is characterized by a wide spectrum of physical features and symptoms that can greatly vary in range and severity. The general physiological status of individuals with WBS included in our study could have negatively affected their adaptive functioning. Furthermore, this effect could have increased with age. These clinical data were not available in our study, so we could not exclude that the worsening in general adaptive functioning could vary according to physical conditions. These considerations could also be applied to individuals with ASD, even though, in this case, clinical data are missing, therefore not allowing us to draw final conclusions. Furthermore, as the final aim of longitudinal studies should be to produce a prediction model on the basis of the data collected over time, and as our study is the first study in this direction, it is not possible yet to assert that what we observed is extendable to all people with WBS and ASD. Further studies are then required to better establish if the results we obtained in our sample can be generalized to general ASD and WBS populations. Finally, we do not have information on the treatment that the children received at T0 and T1, thus making it impossible to distinguish between age and treatment effects. 

To conclude, using a cross-syndrome longitudinal comparison approach revealed partially overlapping profiles in WBS and ASD, despite children with WBS usually being considered as having a better adaptive outcome. Future research should evaluate similarities and differences in all adaptive domains in a more systematic manner by administering homogenous tests and analyzing the possible role of treatment on adaptive functioning. More detailed knowledge about longitudinal trajectories of ASD and WBS could help clinicians to design earlier and more specific interventions. 

## Figures and Tables

**Figure 1 genes-13-01266-f001:**
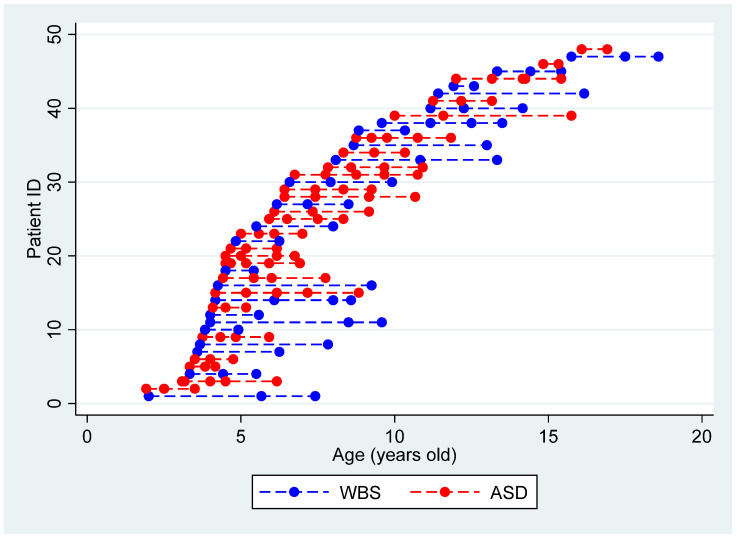
Data collection distribution for each patients’ group.

**Figure 2 genes-13-01266-f002:**
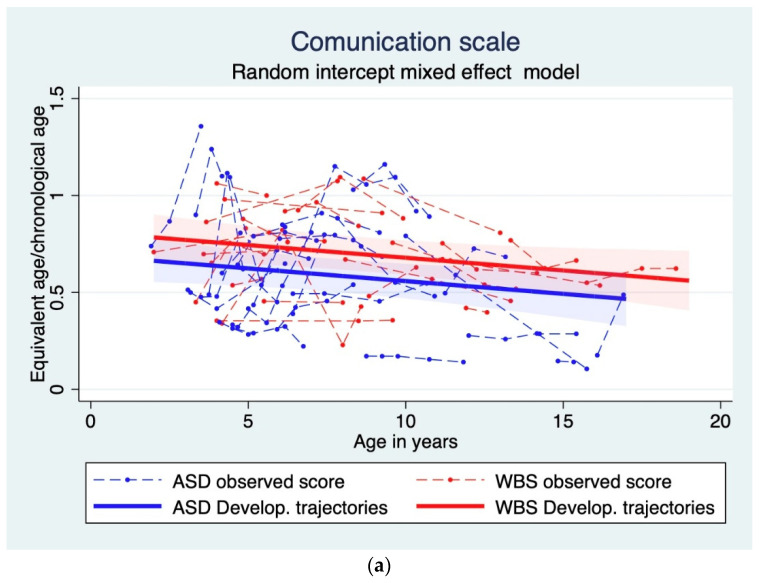
(**a**) VABS communication domain’s developmental trajectories, (**b**) VABS daily living skills domain’s developmental trajectories, (**c**) VABS socialization domain’s developmental trajectories, (**d**) VABS-ABC domain’s developmental trajectories.

**Table 1 genes-13-01266-t001:** Descriptive statistics of ASD and WBS groups.

	ASD	WBS
M: F	3:21	13:11
Age in months (SD) [Min-Max]	80.75 (45.05) (23–193)	82.63 (44.14) (24–189)
IQ/DQ	66.96 (17.12) (39–87)	65.96 (19.24) (39–102)

Legend. M = male; F = female; SD = standard deviation; IQ = intelligent quotient; DQ = developmental quotient.

**Table 2 genes-13-01266-t002:** Comparisons between groups in VABS domains at time TO.

REC	ASD	WBS	*p*-Value
Communication	0.53	0.70	**0.021 ***
Daily Living Skills	0.51	0.55	0.373
Socialization	0.44	0.55	**0.029 ***

Legend: REC, ratio of equivalent age and chronological age; ABC, adaptive behaviour composite; * significant at *p* ≤ 0.05.

**Table 3 genes-13-01266-t003:** Mixed-effect model was fitted to data from the communication, daily living skills, socialization, and composite domains.

Scale	Coefficient (*β*)	*p*-Value	95% CI
**Communication** Age Diagnosis ASD (ref) WBS			
	−0.01	0.033	(−0.025; 0.001)
	0.12	0.055	(−0.002; 0.243)
**Daily Living Skills** Age Diagnosis ASD (ref) WBS			
	−0.02	**<0.001**	(−0.025; 0.009)
	−0.01	0.781	(−0.084; 0.063)
**Socialization** Age Diagnosis ASD (ref) WBS			
	−0.01	**0.005**	(−0.024; −0.004)
	0.12	**0.010**	(0.029; 0.216)
**Total** Age Diagnosis ASD (ref) WBS			
	−0.01	**0.014**	(−0.020; −0.002)
	0.67	0.134	(−0.021; 0.157)

## Data Availability

The raw data supporting the conclusions of this article will be made available by the authors, without undue reservation.

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
