# Peer review of "Differences and Similarities in Adaptive Functioning between Children with Autism Spectrum Disorder and Williams–Beuren Syndrome: A Longitudinal Study"

_genes, 2022, doi:10.3390/genes13071266_

Round 1
Reviewer 1 Report
The current manuscript by Alfieri et al claimed to report a longitudinal cross-syndrome study, comparing adaptive behavior in a sample of children and adolescents with autism spectrum disorder (ASD) and Williams Beuren Syndrome (WBS). This brief report is based on the previous studies carried out by the group. However, it is a bit difficult for this reviewer to follow the track on how this study is unique or important in comparison to other cross-syndrome studies done on ASD and WBS patients or if the authors just try to replicate the previous cross-syndrome studies? They should put more effort into doing a better comparison from the readily available dataset and with their current results to see if they could make a better conclusion. The GWAS mining approach is important as it reports a success that opens a novel avenue for connecting genetics to human physiology. Authors should cite relevant literature and if their contribution on this front is still significant and the application in their short study moves this approach toward a more central activity in the field. The title should be precise and true to the findings. According to this reviewer, the current title is somewhat misleading. The authors produced a prediction model and make a more compelling case for the significance of the identified and should be revised for accuracy.
Reviewer 2 Report
The authors compared longitudinal data of adaptive functioning measured by Vineland Adaptive Behavior Scales (VABS) between two samples of children and adolescents with ASD and WBS, matched for chronological age and cognitive/developmental level at the time of first evaluation. The authors did not find differences on global adaptive level both at first evaluation and over the time. Some differences between individuals with ASD and WBS were detected in Socialization and Communication domains, with children with WBS having a higher level of functioning in both. These conclusions are consistent with literature data. The authors demonstrated that global adaptive functioning ASD and WBS is more similar than thought, even over the time. The impact of having ASD or WBS on adaptive functioning seems to not differ, with both having a similar descending trajectory.
My main doubt concerns the compliance of the article with the profile of the journal Genes. “Genes (ISSN 2073-4425) is an international, peer-reviewed open access journal which provides an advanced forum for studies related to genes, genetics and genomics”. The manuscript contains only brief information on clinical genetic testing in Methods. There is no genetic information in the Results section. The Discussion section contains the only statement relevant to genetics “Genetic basis of similarities and differences in the two groups of patients cannot to date be explained… The etiology of ASD can be linked to different single and multiple chromosomal loci…”. In this context, what is a contribution to the field for the journal Genes?
Major comments
1. The authors incorrectly defined Williams Beuren Syndrome (WBS) as a rare genetic syndrome caused by a de novo homozygous microdeletion on chromosome 7q11.23. The syndrome is associated with a contiguous gene deletion. It has an autosomal dominant pattern of inheritance resulting from the hemizygous deletion (https://www.omim.org/entry/194050?search=Williams%20Beuren%20Syndrome&highlight=beuren%20syndrome%20syndromic%20william).
2. Williams Beuren Syndrome is characterized by a wide spectrum of symptoms and physical features that vary greatly in range and severity. OMIM Clinical Synopsis indicates a variety of body systems that can be affected in persons with WBS. The general physiological status should affect adaptive functioning and this effect can increase with age. The severity of WBS symptoms often varies greatly from case to case (https://rarediseases.org/rare-diseases/williams-syndrome/), however, clinical data are not available and are not taken into account in any way in the statistical analysis.
3. The concern on the severity of the disease also applies to ASD.
Minor comments
1. The manuscript, supplementary and non-published files contain the same tables.
Round 2
Reviewer 1 Report
The authors have addressed my concerns satisfactorily.
Reviewer 2 Report
The manuscript has been improved and can be accepted in the current form.